# Mapping the Oncological Basis Dataset to the Standardized Vocabularies of a Common Data Model: A Feasibility Study

**DOI:** 10.3390/cancers15164059

**Published:** 2023-08-11

**Authors:** Jasmin Carus, Leona Trübe, Philip Szczepanski, Sylvia Nürnberg, Hanna Hees, Stefan Bartels, Alice Nennecke, Frank Ückert, Christopher Gundler

**Affiliations:** 1Institute for Applied Medical Informatics, University Medical Center Hamburg-Eppendorf, 20251 Hamburg, Germany; le.truebe@uke.de (L.T.); sylvia.nuernberg@uk-essen.de (S.N.); h.hees@uke.de (H.H.); f.ueckert@uke.de (F.Ü.); 2Hamburg Cancer Registry, Authority for Science, Research, Equality, and Districts, 20097 Hamburg, Germany; philip.szczepanski@bwfgb.hamburg.de (P.S.); alice.nennecke@bwfgb.hamburg.de (A.N.); 3University Cancer Center Hamburg, University Medical Center Hamburg-Eppendorf, 20251 Hamburg, Germany; st.bartels@uke.de

**Keywords:** cancer registry, standardized vocabulary, semantic interoperability, common data model, OMOP

## Abstract

**Simple Summary:**

Resident physicians and medical institutions in Germany are required to report diagnostics, treatments, progression, and follow-up information for tumor patients to the respective state cancer registries. The information is transmitted electronically according to a defined data scheme (oncological basis dataset [oBDS]). In this study, we first mapped oBDS elements to the standardized vocabularies, a metadata repository of the observational medical outcomes partnership (OMOP) common data model (CDM). The mapping of the oBDS to the standardized vocabularies promotes the semantic interoperability of oncological data in Germany and provides the opportunity to participate in network studies of observational health data sciences and informatics under the usage of federated analysis.

**Abstract:**

In their joint effort against cancer, all involved parties within the German healthcare system are obligated to report diagnostics, treatments, progression, and follow-up information for tumor patients to the respective cancer registries. Given the federal structure of Germany, the oncological basis dataset (oBDS) operates as the legally required national standard for oncological reporting. Unfortunately, the usage of various documentation software solutions leads to semantic and technical heterogeneity of the data, complicating the establishment of research networks and collective data analysis. Within this feasibility study, we evaluated the transferability of all oBDS characteristics to the standardized vocabularies, a metadata repository of the observational medical outcomes partnership (OMOP) common data model (CDM). A total of 17,844 oBDS expressions were mapped automatically or manually to standardized concepts of the OMOP CDM. In a second step, we converted real patient data retrieved from the Hamburg Cancer Registry to the new terminologies. Given our pipeline, we transformed 1773.373 cancer-related data elements to the OMOP CDM. The mapping of the oBDS to the standardized vocabularies of the OMOP CDM promotes the semantic interoperability of oncological data in Germany. Moreover, it allows the participation in network studies of the observational health data sciences and informatics under the usage of federated analysis beyond the level of individual countries.

## 1. Introduction

Cancer continues to be a major challenge of modern societies. Approximately 240,000 people are expected to die of cancer in the year 2023 in Germany alone [1]. To assure, improve, and advance the quality of oncological care and to provide a basis for future clinical research, uniform clinical documentation of oncological data is necessary. The Cancer Screening and Registry Act (*Krebsfrüherkennungs- und -registergesetz* [KFRG], in German) came into force in Germany in 2013. It builds the legal basis for a uniform nationwide clinical cancer registration by all federal states in Germany. Physicians or institutions like treatment centers or hospitals are statutorily obligated to report diagnostics, treatments, progression, and follow-up information for cancer patients to the respective cancer registry. Thereupon, the Working Group of German Tumor Centers (*Arbeitsgemeinschaft deutscher Tumorzentren e.V.* [ADT]) and the Association of Population-Based Cancer Registries (*Gesellschaft der epidemiologischen Krebsregister in Deutschland e.V.* [GEKID]) developed the uniform oncological basis dataset (*onkologischer Basisdatensatz* [oBDS], in German) in 2014 [2]. Oncological data are recorded by all 15 federal state cancer registries in Germany according to the oBDS. The reporting includes all malignant tumor entities, in situ stages of malignant neoplasms, and benign tumors of the central nervous system (C and D prefixes within the International Statistical Classification of Diseases and Related Health Problems, 10th revision, German Modification [ICD-10-GM]) [3,4].

Over the past few years, continuously more research studies evaluating these cancer registry data have been published as the data quality and analysis possibilities have improved [5]. To give an example, a population-based cohort study by Gennari et al. investigated the impact of the type of surgical method (minimally invasive vs. open) on the survival of patients with early-stage cervical cancer. A prognostic covariate analysis showed that the surgical approach was not associated with the survival of patients but rather with the certification status of the treatment center [6]. Building on similar findings, Cheng et al., investigated the cost-effectiveness of cancer care in certified compared to noncertified hospitals. It was demonstrated that not only the survival of patients treated in certified hospitals was longer, but also the costs were lower in comparison to those of noncertified hospitals [7,8].

With a rise in complex diagnostic approaches and treatment options in oncology and the shift towards personalized medicine [9], the amount and complexity of data to be recorded have increased simultaneously. While the oBDS is continuously reviewed and extended, data merging and processing for nationwide research projects remain a challenge. To exploit the full potential of the recorded oncological data, the German parliament updated the Federal Cancer Registry Act (*Bundeskrebsregisterdatengesetz*, [BKG], in German) in 2021, aiming to combine both population-based and treatment-related data from all federal state cancer registries in Germany at the Center for Cancer Registry Data (*Zentrum für Krebsregisterdaten* [ZfKD]) of the Robert Koch Institute [10]. The introduction of the oBDS in 2014 and the implementation of XML interfaces has already created the required syntactic interoperability for this purpose. Nevertheless, the usage of different documentation software solutions across the reporting institutions, for example Tristan or the Giessener Tumor Documentation System (GTDS), has led to semantic and technical heterogeneity [11]. The involved cancer registries hence fulfill the prerequisites to communicate and exchange data regardless of the use of different systems (syntactic interoperability) but lack the correct interpretation due to ambiguous semantics. Consequently, the oBDS information is partly processed differently and stored in separate backend architectures. Major preparation and processing of the XML packages to correct possible transmission errors and to ensure better data quality is necessary prior to data export to enable joint analysis or data combination at the ZfKD. This is not only time-consuming but also complicated due to the missing semantic interoperability. The 2021 Cancer Registry Data Consolidation Act (*Gesetz zur Zusammenführung von Krebsregisterdaten*, in German) addresses precisely this problem. The aim is to investigate how the various semantic differences in the processing of reported cancer registry data can be resolved to better utilize these oncological data in a scientific context. A possible approach to bridge the semantic heterogeneity of different data documentation systems could be the transfer of the oBDS to a Common Data Model (CDM). A CDM represents unified information and enables decentralized analysis. Data protection can thus be guaranteed as the data are stored locally, and aggregated results are only exchanged via common analysis scripts. To realize this, the data must be transformed into a standardized format defined by entities, attributes, and relationships. This enables data comparability within the CDM, despite the integration of different operational data sources. In a systematic review, Pardee and Weeks investigated and presented several CDMs that cover the fields of clinical and collaborative research [12]. In a study by Garza et al. some of the CDMs introduced by Pardee and Weeks (Sentinel Common Data Model [SCDM] v.5.0, Patient-Centered Clinical Research Network [PCORnet] v.3.0, Observational Medical Outcomes Partnership Model [OMOP] CDM v.3.0, Data Interchange Standards Consortium [CDISC], Study Data Tabulation Model [SDTM] v.1.4.) were evaluated in terms of their completeness, integrity, flexibility, integrability, and implementability for electronic health record (EHR)-based longitudinal registry data. OMOP CDM v3.0 met most of their evaluation criteria [13].

The data used in this feasibility study were derived from the Hamburg Cancer Registry (*Hamburgisches Krebsregister* [HKR]). The HKR collects information according to the oBDS for all cancer patients from the time of new diagnosis until death by all reporting institutions in the federal state of Hamburg. The collected information is further processed by the HKR for quality assurance and then transferred for research purposes to the ZfKD.

Due to the uniform semantic processing of the data within the OMOP CDM, the transfer of the oBDS to the standardized vocabularies within the OMOP CDM leads to a high level of semantic interoperability, which means that oBDS data can also be easily incorporated into network studies in an international context while maintaining local data protection. This feasibility study focused on the extent to which the oBDS characteristics can be represented by the standardized vocabularies of the OMOP CDM. Therefore, (1) the preparation of the metadata is displayed. This can be performed automatically using the standardized vocabularies provided by the OHDSI community or manually. Metadata preparation, which includes the look-up tables for the automatically created mappings and the manually created ones, is essential for the (2) extract transform load process, which semantically annotates the patient source data while incorporating the generated metadata that adhere to OMOP CDM conventions. Building on this, we document the process and (3) outcome of mapping the HKR patient data and reporting centric-source data to the new terminologies.

## 2. Materials and Methods

The implementation of the OMOP CDM at the HKR required a series of tasks which are schematically summarized in Figure 1. The analysis of the oBDS and HKR source data (Figure 1a) was followed by the mapping of the oBDS metadata to the standardized vocabularies of the OMOP CDM (Figure 1b). In a final step, an Extract–Transform–Load (ETL) process was applied for semantic annotation of the HKR source data to the new terminologies (Figure 1c).

### 2.1. oBDS Source Data

The oBDS (version 2014, previously referred to as *ADT/GEKID-Basisdatensatz*) can be divided into 20 categories. These categories can be assigned to items with a defined, finite, or infinite number of expressions (free text). In the category diagnoses, for example, the ICD-10-GM is stored as an item which, in turn, allows a defined number of expressions from the ICD-10-GM with a C or D prefix. For other items, such as information on previous tumor disease or the administration of substances included in the systemic therapy category, the information is submitted in the form of free text, resulting in an infinite number of expressions. Other items, such as post-interventional surgical complications, are based on a German surgical complication standard which was introduced together with the oBDS release (e.g., ANI = acute renal failure, *Akute Niereninsuffizenz*).

All oBDS terminologies were manually translated into English with DeepL (www.deepl.com (accessed on 31 May 2023)), or in the case of larger data standards, the translation was automated using Python 3.9.13 and the googletrans library [14,15].

### 2.2. HKR Source Data

The cohort was phenotyped as follows: all patients with a malignant primary tumor of the lung or breast carcinoma entities (C34 and C50, according to the ICD-10-GM) who were first diagnosed in 2016 or later. The dataset included all items defined by the oBDS.

All information about a cancer patient collected by the HKR is linked to the reporting institution. Therefore, information collected for one patient can be reported by several institutions. Consequently, the information can primarily be used for institutional statistics, but its use in the clinical context is limited due to double reporting. The OMOP CDM is a patient-centric relational data model, and thus, data processing was required for each record prior to OMOP CDM implementation. The data were continuously processed and merged by the HKR according to defined rules and transferred to flat files representing the clinically best available and correct patient-centric data (the so-called “best-of”).

In summary, the following flat files were extracted from the HKR: (1) twenty-three flat files reporting institution-centric information, (2) eleven best-of flat files with patient-centric information, and (3) seven look-up flat files containing all references (e.g., common toxicity criteria). The content from (2) and (3) cover almost the entire oBDS and were therefore used in the context of this work. Missing information was supplemented using (1) [3]. For reasons of data protection, all patient- and reporting-institution-identifying information was excluded.

### 2.3. Mapping oBDS Metadata to the Standardized Vocabularies of the OMOP CDM

OMOP CDM version 5.4 was used for this feasibility study. Crucial for the OMOP CDM implementation are the standardized vocabularies, a common repository developed and maintained by the Observational Health Sciences and Informatics (OHDSI) community. They ensure the standardization of the local source data during the ETL to comply with the OMOP CDM conventions. The standardized vocabularies are organized into domains, such as the *drug* or *measurement* domains, and vocabularies representing sets of concepts from existing data standards. In total, the standardized vocabularies currently cover 150 data standards, ranging from disease definitions (e.g., SNOMED or ICD-10-GM) to procedures (e.g., *Operationen- und Prozedurenschlüssel* [OPS]), drug standards (Anatomical Therapeutic Chemical [ATC], RxNorm), and genetic vocabularies (e.g., Catalogue Of Somatic Mutations In Cancer, [COSMIC]) [16].

All local codes of the oBDS must be linked to the equivalent standardized vocabulary of the OMOP CDM. Dependent on the processing efforts, the oBDS items are categorized into (1) automatic mapping, (2) manual mapping, (3) ETL, and (4) not mappable (Figure 1b). The collection of all oBDS items is referred to as metadata. Appendix A lists all required data standards for OMOP CDM implementation.

For automatic mapping, the OMOP CDM CONCEPT_RELATIONSHIP table was used to generate the source_to_standard table. The latter was applied during the ETL process for semantic annotation (Figure 1c). This look-up table is essential for the ETL, and within it, it is used as a basis for the automatic mapping process of the local codes of the oBDS to standard concepts within the standardized vocabularies. Some oBDS items require further data preparation to achieve automatic mapping, especially items with infinite expressions in the form of free text. To reach a certain data quality standard, the substance information was cleaned using regular expressions. This allowed, for example, automatic mapping on RxNorm, which is the respective data standard for the OMOP CDM *drug* domain [17]. Intricate free-text expressions can only be processed by RegEx to a limited extent. Examples are the oBDS treatment protocols which were mapped to the OMOP CDM treatment regiments. Therefore, the OncoRegimenFinder algorithm, established by the Oncology Workgroup of the OHDSI community, was applied. This algorithm summarizes all substances administered to a patient within the last 30 days and maps them to the correlating *HemOnc* concept, a data standard used to represent treatment pathways [18].

Local data standards that were not covered by the data standards included in the standardized vocabularies required manual mapping to the OMOP CDM. The graphical interface Athena was used to associate the same or a similar semantic concept within the standardized vocabularies to the local codes [16]. The software Usagi v.1.4.3 was applied to map larger local data standards to the standardized vocabularies. It follows the term frequency inverse document frequency (TF-IDF) approach in the field of natural language processing (NLP) [19,20]. Both types of software were used to generate the source_to_concept table which was applied during the ETL process for semantic annotation [21]. Because the source_to_concept look-up table must be created manually, whereas the source_to_standard look-up table is created automatically under the usage of the OMOP CDM CONCEPT_RELATIONSHIP table, the creation of the source_to_concept table takes more time. This look-up table is essential for mapping the local codes of the oBDS that are not covered by the standardized vocabularies (e.g., surgery codes) within the ETL.

Items, such as dates in the oBDS, are not mapped in a previous step via the source_to_standard or source_to_concept look-up tables, and their assignment to the corresponding OMOP target column is performed directly during ETL. An overview of the assignment of the corresponding oBDS items and their mapping approach can be found in Appendix A. It should be noted that not all expressions of the oBDS items can always be mapped to a standard concept of the standardized vocabularies. Therefore, it is possible that only some of the expressions of one oBDS item were mappable, whereas others within these items were still not mappable. This occurred, for example, when specifying the side localization of the tumor. The expressions right (oBDS Expression = “R” [Concept_id: 36770058]), left (oBDS Expression: “L” [Concept_id: 36770232]) and midline/center (oBDS Expression = “M” [Concept_id=36770562]) can be mapped within the Cancer Modifier Vocabulary of the standardized vocabularies, but the expressions unknown (oBDS Expression = “U”) and both sides (oBDS Expression =“B”) cannot be displayed via the standardized vocabularies.

Items categorized as not mappable are discussed below.

### 2.4. ETL

During the ETL process, the HKR-source data were transformed and loaded to the new terminologies of the OMOP CDM for semantic annotation (Figure 1c). The OMOP CDM infrastructure was established on an external server in compliance with data protection regulations.

A PostgreSQL database was set up, and an import script was written to transfer all flat files into source tables within the relational PostgreSQL database design (Figure 2, import). For more complex transformations, a stage level was implemented following the Kimballs and Caseras guidelines for the design of a data warehouse [22] (Figure 2, stage).

OMOP is a patient-centric model that focuses on patients who are connected to hospitals as part of their treatment. Therefore, it requires a unique visit ID (visit_occurrence_id) and a date for each clinical event during a patient’s visit in a specific hospital [21]. This link is not depicted within the cancer registry dataset. Therefore, the data were filtered prior to the ETL for dates correlating to each clinical event via predefined rules and assigned to a visit accordingly. Results were collected in the source table Eventliste. This table is necessary in the ETL for assigning clinical events to specific visits, as it is required for OMOP implementation.

The target tables were created in the PostgreSQL data warehouse structure in the cdm schema (Figure 2, cdm). For automation of the ETL process, batch files were written, and their execution was controlled by the Windows Task Scheduler. Table 1 lists all source tables used during the ETL to populate the target tables of the OMOP CDM. After successful implementation of the ETL process, the Achilles tool was used to check for logical errors [23].

### 2.5. Data Analytics and Visualization

To compare the consistency among the oBDS, the HKR-source data, and the content mapped to the OMOP CDM, R 4.1.2, Phyton 3.9.13, and SQL were used for the data analysis. R 4.1.2 and Phyton 3.9.13 were used for data visualization. Microsoft PowerPoint v.2019 was used for the remaining visualizations.

## 3. Results

The oBDS represents the basis for uniform, nationwide clinical cancer registration by all federal states in Germany. It requires information regarding the diagnostics, treatments, and progression and follow-up information about the patient’s cancer. This feasibility study examined the extent to which the oBDS data elements can be represented by the OMOP CDM to acquire a standardized format. As a proof-of-principle study, the cancer registry data from the HKR were transferred to new terminologies of the OMOP CDM. This section provides insights about the output from the metadata mapping.

### 3.1. Metadata Mapping

A total of 17,840 oBDS expressions were examined regarding their mappability to a standard concept within the standardized vocabularies of the OMOP CDM. Table 2 provides an overview of 19 oBDS categories, items, and expressions. The 20th oBDS category *reporting reason*, which specifies the details of the patient’s declaration of consent, was not considered for further investigation.

A comprehensive mapping overview is provided in Appendix A. It illustrates the mapping flow from the oBDS category to the target vocabulary within the OMOP CDM.

In total 17,744 (99.46%) oBDS expressions could be represented via standard OMOP CDM concepts. Of these, 16,946 (94.99%) were automatically mapped via the CONCEPT_RELATIONSHIP table to the source_to_standard table, 746 (4.18%) were manually assigned to and implemented in the source_to_concept table, and 52 (0.29%) expressions were mapped during the ETL process. ETL mapping mainly concerns date entries in the oBDS. Figure 3 shows the proportion of each mapping status for all oBDS expressions per oBDS category. The *TNM classification* category contains the most elements that cannot be displayed by a standard concept within the standardized vocabularies (not mappable: 76.4%). Other categories, such as *patient master data* and *register master data*, contain person- or register-identifying information, and were therefore categorized as not mappable. Because of data protection regulations, this information is not part of the OMOP CDM. However, most of the oBDS data elements within the categories *death, diagnosis, histology, surgery*, and *systemic therapy side effects* could be transferred to standard concepts of the OMOP CDM (not mappable: <2.4%).

Due to the ontological structure of the standardized vocabularies, standard and non-standard OMOP CDM concepts are connected to other concepts via relationship_IDs. Thus, additional information about each concept can be easily retrieved. With the mapping of the 17,744 oBDS expressions to standardized OMOP CDM concepts, a total of 380,343 relationships were queried. The most common relationships are shown in Figure 4 with the highest values for *subsumes* (*n* = 198,089; 52.1%), *mapped from* (*n* = 59,342; 15.6%), and *asso morph of* (*n* = 182,283; 8.5%), whereas the *value to schema* relationship (*n* = 5; 0%) is the least present. In addition, the oBDS categories *surgery* (*n* = 198,496; 52.2%), *diagnosis* (*n* = 134,834; 34.5%), and *histology* (*n* = 24,200; 6.4%) are best embedded within the OMOP CDM.

### 3.2. ETL

The HKR-source data were transformed and loaded into the OMOP CDM during the ETL process. A total of 33,432 patients could be transferred from the HKR database to the OMOP CDM. This corresponds to a total mapping rate of 100% of all patients recorded. Table 3 summarizes the absolute number of recorded patient information per oBDS item. Only those items whose information was obtained from best-of flat files were analyzed. Items only obtained from reported flat files were excluded from this analysis. The following patient-centric oBDS items could be transferred to the OMOP CDM with a mapping rate of 100% (Table 3): topography, condition accuracy, ICD-10-GM, overall residual tumor status, local residual tumor status, morphology ICDO, metastasis, CTCAE, substances, other classification, ECOG + Karnofsky, former cancer diseases, topography site, and ICDO3. The following items could be transferred with a mapping rate of below 50%: nodes (mapping rate: 33.64%), sentinel nodes (mapping rate: 22.92%). The inclusion of information from the reference lists in the standardized vocabularies of the OMOP CDM can be resulted in a 1:n map, leading to a higher number of entries in the target table than in source data. This applies to the item OPS with a resulting mapping rate of 140.51%.

### 3.3. Oncological Representation via OMOP CDM

By implementing the EPISODEtable within the OMOP CDM, a total of 171,036 cancer-related episodes were recorded. The episode describing *surgical interventions,* assigned to the *treatment* class within the standardized vocabularies, was the most frequently transcribed as cancer-related episode with 47,912 entries. In total, 115,478 treatment-related episodes were derived from the HKR-source data. A total of 5969 episodes were converted to the *disease dynamic* concept class, which displays the individual remission status of a patient after treatment. The *concept of progression,* with a total of 4834 entries, was the most frequently implemented. For the determination of the *disease extent*, the *TNM* data were derived from the best-of flat file containing patient treatment information. We were able to derive 14,832 entries from the HKR-source data.

By applying the OncoRegimenFinder algorithm (see Appendix A), the substances administered to a patient within the last 30 days were mapped to the OMOP treatment regiments represented via the *HemOnc* data standard within the standardized vocabularies. A total of 13,518 entries were available within the HKR source data with treatment information provided in form of free text. In total, 8419 treatment regimens could be derived via OncoRegimenFinder algorithm. This corresponds to a mapping rate of 62.28%. Since treatment regimens are a combination of drug data, a mapping comparison with drug information was impossible.

## 4. Discussion

With this feasibility study we were able to show that most of the oBDS characteristics can be displayed via standard concepts within the standardized vocabularies of the OMOP CDM. Nevertheless, there are major differences in the mappability of individual elements within the oBDS categories. In particular, oBDS master data elements containing detailed information about the reporting institutions and patients exhibit gaps regarding their representability via standard concepts of the OMOP CDM. The primary intended use of the OMOP CDM is the design and execution of international network studies. To overcome issues with data protection regulations in these kinds of studies, the recording of sensitive personal data is not included in the OMOP CDM. However, parameters of patient master data related to the phenotyping of cohorts in oncological research, such as the date of birth, sex, and country of birth, are queryable within the OMOP CDM. All elements related to the oBDS register of master data containing information about the reporting institutions were categorized as not mappable by default. The OMOP is a patient-centric data model that does not primarily provide information about the reporting institution in the context of cancer transmission. Future investigations should focus on the extent to which it is possible to enable the integration of the reporting logic of cancer registrations in Germany into the OMOP CDM.

In addition, the oBDS contains elements that cannot be mapped to the standardized vocabularies of the OMOP CDM but are of high interest for oncological research. The tumor (T), nodes (N), and metastasis (M) classification data should be viewed as particularly critical in this context. Although both clinical and pathological T, N, and M indications can be represented using the standardized vocabularies, another large part of the oBDS TNM classification data cannot be represented. The oBDS provides additional TNM information, for example, if the TNM classification is performed during or after initial multimodal therapy, if it is a recurrence classification or primary tumor, if the TNM classification was conducted in the context of an autopsy, if multiple primary tumors are present, and if there is a lymphatic, perineural, or venous invasion or if the serum tumor markers are elevated. Currently, this additional TNM information cannot be displayed via the standardized vocabularies of the OMOP CDM. We should therefore review whether this information can be added as a novel concept to the standardized vocabularies in the future.

However, regardless of the slight loss of information, the data comparability improved. In a previous study, we were able to show that the survival probability of patients with breast carcinoma calculated via the source system and the OMOP CDM does not differ significantly and that the OMOP CDM is a sufficient tool for data analysis in the context of oncological data from Germany [24]. In addition, analyses can be performed in a decentralized manner. For example, an analysis script can be executed at different locations, and the aggregated results can subsequently be compared. This could be particularly interesting in countries with high data protection regulations.

However, any use case should be examined for its required OMOP CDM display and mapping options before starting the implementation of the system. If it is feasible, the OMOP CDM is a powerful tool to create high semantic interoperability. The uniform presentation of clinical information via the OMOP CDM can considerably minimize the data preparation and processing workflow within the framework of a multicentered study. In particular, during the COVID-19 pandemic, its flexibility and speed were demonstrated to benefit evidence-based research by the OHDSI research community. In October 2020, a study was published that phenotyped more than 34,000 COVID-19 hospitalized patients from Asia, Europe, and North-America and compared them to the characteristics of influenza cohorts [25]. Another study from the OHDSI community investigated the hospitalization characteristics of obese compared to non-obese COVID-19 patients [26]. The OHDSI community has also developed prediction and analysis tools in the field of methodological research, for example, regarding model validation in artificial intelligence. In a study by Reps et al., the research network is actively used for the external validation of machine learning models [27]. Moreover, in the field of cancer research, the OHDSI network is delivering evidence-based results. Cancer-related studies can focus on methodological aspects, such as the recruitment of study participants in clinical trials. On the other hand, they can focus on the investigation of relevant cancer outcomes or the individual risk assessment of a patient in observational network studies, for example, the probability of developing a secondary malignant neoplasm after treatment with radioiodine for thyroid cancer [28,29,30]. Mapping the oBDS dataset to the standardized vocabularies of the OMOP CDM could enable comprehensive analyses among German cancer registries. It also offers the possibility of conducting studies with transnational cancer registries, such as the SEER cancer registries from the USA, in compliance with the current data protection regulations.

### 4.1. Limitations

#### 4.1.1. Metadata

It is difficult to map the oBDS elements transmitted in the form of free text to a standard concept within the standardized vocabularies of OMOP due to a lack of standardization. This particularly applies to the transmission of substance information and treatment protocols/regimens. In this study, the substance data transmitted to the HKR were cleaned using regular expressions and then mapped to the ATC data standard. This process was accompanied by a loss of data, as not all substance data could be mapped to a corresponding ATC concept. The treatment regimens were also submitted in the form of free text. However, the discrepancies in the transmitted reports are even more heterogeneous than in the case of substance transmissions. Here, detection with regular expressions was not possible. It was therefore decided to use the Oncology Workgroup’s OncoRegimenFinder algorithm, developed by the OHDSI community [18]. This algorithm summarizes all administered substances within the last 30 days and examines whether a suitable *HemOnc* concept can be found for the derived substance combinations. For all cancer registries looking to use the metadata as part of their OMOP CDM implementation, an application of the OncoRegimenFinder is required to assign the data entries to the oBDS concept class regimens.

All mappings included in this feasibility study were carried out by a single person. However, according to official recommendations, this should be carried out by an interdisciplinary group of experts to achieve a high mapping quality that reflects the different facets of reality. Furthermore, it must be noted that manual mapping has not been further validated. This was considered at the beginning of this work but was not pursued further due to a lack of personnel and time. Validation of a small, randomly selected sample would have been good here.

#### 4.1.2. ETL

The entire ETL process was also written by one person only. After completion, the content of this process was checked for errors using the Achilles tool, and the source system was continuously compared with the OMOP CDM using our own developed unit tests [23]. The occurrence of logical errors during an ETL process cannot be ruled out. Therefore, it is important for scientists to report errors to the OMOP CDM development team, so that improvements to the ETL can help to avoid these mistakes in the future.

Furthermore, only regular expressions were used for the free-text statements within the oBDS. However, natural language approaches have continuously been improved and developed further. It can be assumed that a combination of both methodological approaches can lead to an improved mapping rate for mapping the substance name to the ATC.

It should additionally be considered that it is difficult to map reported institution-centric data to a patient-centric model while transferring the correct content. The data acquisition of individual patients at several institutions results in multiple reports. Nevertheless, an attempt was made to eliminate duplicates of patient data within the reporting institution-centric flat files by joining them with patient-centric best-of flat files. As soon as all remaining best-of flat files are defined within the framework of platform § 65c, the nationwide association of clinical cancer registries, the reporting institution-centric flat files currently used in the ETL should be replaced.

### 4.2. Lessons Learned

The greatest strength of the OMOP model lies in the area of network studies. The development and implementation of network study protocols is based on the OMOP data model and is technically easy to implement (after OMOP CDM implementation) while maintaining local data protection. Nevertheless, source data information is getting lost during the mapping process, since oBDS codes are often more granular here as it can be represented using the standard concepts of the standardized vocabularies. Therefore, the use case and the investigation of whether this use case is feasible should be investigated beforehand. In addition, it was found that, for some tasks in the context of OMOP implementation, working together in a team saves time and probably leads to a better result, e.g., manual mapping of the oBDS codes. This should be considered in future work. In addition, it should be determined who will continue to administer the OMOP CDM after successful implementation, since it has to be maintained and provided with regular updates (e.g., vocabulary updates from the OHDSI).

A last point we have taken from this project is that this project serves as a starting point for us. This study investigated the oBDS in its initial version from 2014. For the future work, we would like to investigate the extent to which the genetic vocabulary introduced with the oBDS 3.0.0 can be represented via the OMOP CDM. If this is possible, the OMOP model may represent a significant step toward a personalized medicine approach to patient oncology care.

## 5. Conclusions

This study shows that a large portion of the oBDS characteristics can be represented by standard concepts contained within the OMOP CMD. Nevertheless, the use case for an OMOP implementation must be clearly defined and delimited. The OMOP CDM is a powerful tool, especially for the achievement of high semantic interoperability among cancer registries in Germany. This will lead to comparability of the data and will enable the inclusion of the existing analysis infrastructure of the OHDSI community. Cross-registry analyses in compliance with current data protection legislation in Germany could easily be realized with the OMOP CDM. However, best-of information must be available to accomplish the correct mapping to the OMOP CDM. Of note are data reported on systemic and radiation therapy which, until now, have generally lacked best-of information. In addition, there are gaps in the representation of reporting processes in the OMOP CDM, as the tool follows a patient-centric approach. The implementation of a reporting logic was not considered in this study. If deemed relevant for German cancer registries, this should be further investigated. Furthermore, it should be noted that free-text information mappings lead to a loss of information. In addition to being technically difficult to apply within the ETL, it creates a massive performance loss due to complex transformations. It is therefore recommended that future oBDS updates should eliminate free-text options, especially for clinically relevant information (e.g., substance administration in the context of chemotherapy, chemotherapy protocols). To fully support an easy OMOP CDM implementation, the oBDS should be part of the standardized vocabularies to avoid manual mapping of the source data as preparatory work for the ETL process. In summary, this work aims to serve as a starting point to further advancing the data harmonization of cancer registry data in Germany.

## Figures and Tables

**Figure 1 cancers-15-04059-f001:**
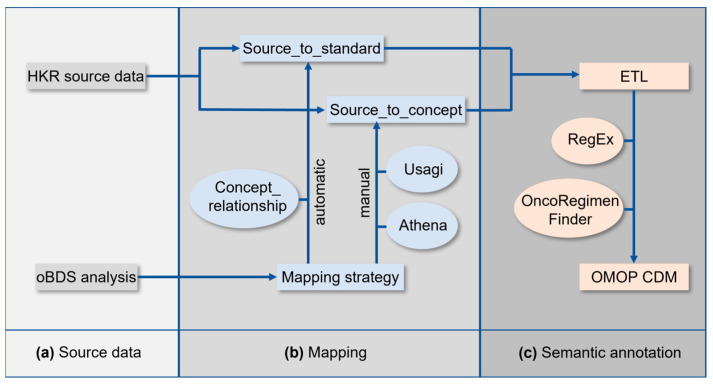
Schematic approach of OMOP implementation at the Hamburg State Cancer Registry (HKR). (**a**) Pre-analysis of the source data and the oncological basis dataset (oBDS), (**b**) mapping, and (**c**) semantic annotation.

**Figure 2 cancers-15-04059-f002:**
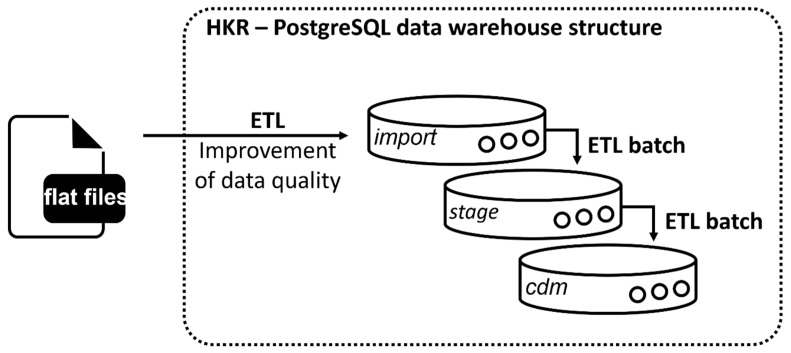
HKR data warehouse structure used to transform the HKR source data into the OMOP CDM format using a PostgreSQL v.14 architecture. Automatization during the ETL was conducted by batch processing.

**Figure 3 cancers-15-04059-f003:**
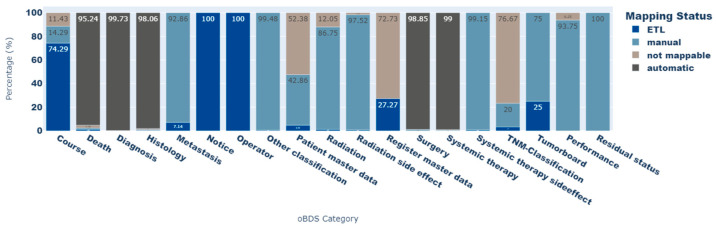
Proportions of the mapping statuses of oBDS expressions depending on their oBDS categories.

**Figure 4 cancers-15-04059-f004:**
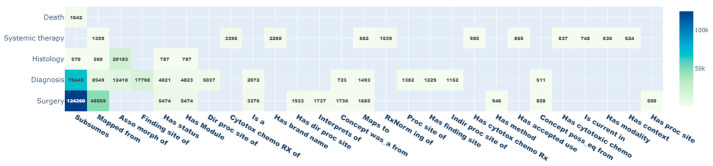
Heatmap of the ontological integration of the oBDS metadata into the standardized vocabularies (*n* ≥ 500).

**Table 1 cancers-15-04059-t001:** Tables used in the ETL process and the number of rows per OMOP CDM target table. Italics: tables from the OMOP model or generated via OHDSI tools, capital letters: HKR tables, others: tables created as part of the ETL process.

OMOP CDM Target Table	Number of Rows [*n*]	HKR Source Tables
Person	33,432	PATIENT
Death	2831	MELDUNG, MELDUNGVM, Eventliste
Visit_occurrence	512,824	Eventliste
Device_exposure	8	BESTOFTHERAPIEOPS, Eventliste
Condition_occurrence	98,971	Eventliste, BESTOFTHERAPIENEBENWIRKUNG, BESTOFTHERAPIEKOMPLIKATION, BESTOFDIAGNOSE, MELDUNGMODULMAMMA, MELDUNG
Drug_exposure	30,158	BESTOFTHERAPIESUBSTANZ, BESTOFTHERAPIE, ATC, Eventliste
Procedure_occurrence	156,823	MEDLUNGTMSYART, MELDUNG, Eventliste, BESTOFTHERAPIEOPS, MELDUNGTMSTBESTRAHLUNG, BESTOFTHERAPIE, MELDUNGTM, MELDUNGMODULMAMMA
Measurement	623,335	bestoftuk, Eventliste, BESTOFDIAGNOSE, BESTOFDIAGNOSEMETASTASEN, BESTOFDIAGNOSEKLASSIFIKATION, MELDUNGMODULMAMMA, MELDUNG, BESTOFTHERAPIEOPS
Observation	178,494	BESTOFDIAGNOSE, Eventliste, MELDUNGTMSYART, MELDUNG, BESTOFTHERAPIE, BESTOFTHERAPIEOPS
Episode	136,497	*Condition_Occurrence*, BESTOFFOLGEEREIGNIS, BESTOFDIAGNOSE, BESTOFTHERASPIEOP, BESTOFTHERAPIE, *regimen_ingredients, drug_exposure,* Eventliste, *Procedure_occurrence*
Episode_event	2180.903	*Episode, Condition_Occurrence,* *Measurement, Procedure_occurrence, Drug_Exposure, Observation*

**Table 2 cancers-15-04059-t002:** oBDS categories with the number of items present and their numbers of expression.

oBDS Category	Items [*n*]	Expressions [*n*]
Course	5	35
Death	3	84
Diagnosis	10	5929
Histology	10	1080
Metastasis	2	14
Notice	1	1
Operator	1	1
Other classification	5	191
Patient master data	15	21
Performance	1	16
Radiation	10	166
Radiation side effect	3	121
Register master data	11	11
Residual status	2	13
Surgery	5	7414
Systemic therapy	8	2591
Systemic therapy side effect	3	118
Tumor (T), node (N), metastasis (M) classification	16	30
Tumor board	2	4
	**113**	**17,840**

**Table 3 cancers-15-04059-t003:** The number of patient records that could be transferred from the source system to the OMOP CDM divided by domain. In addition, the tables show the mapping rates of the source data to the target system divided by the domain.

oBDS Item	OMOP Domain	HKR [*n*]	OMOP [*n*]	Mapping Rate [%]
Topography	Measurement	34,343	34,310	99.90
Nodes	Measurement	13,654	4593	33.64
Treatment intention	Procedure	44,574	42,126	94.51
Condition accuracy	Measurement	31,940	31,940	100
ICD-10-German Modification	Condition	34,405	34,405	100
Grading	Measurement	26,993	25,565	94.71
Overall residual tumor status	Observation	223	223	100
Local residual tumor status	Observation	20,879	20,879	100
cN or pN	Observation	32,056	26,933	84.02
Morphology ICDO	Observation	34,321	34,321	100
Metastasis	Measurement	15,030	15,030	100
CTCAE	Condition	3489	3489	100
Substances	Drug	30,118	27,140	90.11
Treatment complication	Condition	771	745	96.63
Other classification	Measurement	1655	1655	100
cM or pM	Observation	32,056	26,188	81.69
ECOG + Karnofsky	Observation	7928	7928	100
Former cancer diseases	Condition	228	227	99.56
Topography site	Measurement	31,163	31,163	100
OPS	Procedure	29,924	42,046	140.51
ICDO3	Condition	34,405	34,405	100
cT or pT	Observation	32,056	27,311	85.20
Sentinel nodes	Measurement	9444	2165	22.92

## Data Availability

Restrictions apply to the availability of these data. Data were obtained from the federal state cancer registry during the funding of the project. A data use request can be submitted to the Hamburg Cancer Registry.

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
