# Peer review of "Mapping the Oncological Basis Dataset to the Standardized Vocabularies of a Common Data Model: A Feasibility Study"

_cancers, 2023, doi:10.3390/cancers15164059_

Round 1

Reviewer 1 Report

I would like to express my appreciation to the authors for their significant work in addressing the semantic and technical heterogeneity of oncological data in Germany. As someone who has worked with cancer registries, I understand the challenges associated with data harmonization and terminology concerns, making this study particularly relevant.

While the study provides valuable insights into the mapping of the oncological basic dataset (oBDS) to standardized vocabularies, I found the content somewhat difficult to follow. I would recommend the authors simplify the presentation of their findings, utilizing clearer language and a more structured approach. 

Additionally, I believe incorporating manual validation alongside the automatic mapping process would be beneficial for at least a small amount of randomly selected data.

It would also be valuable if the authors could provide further details regarding the feasibility of the mapping process, such as cost, time, and computational power requirements. 

Author Response

Dear Reviewer,

please see our response to your very valuable comments in the attachment.

Reviewer 2 Report

The paper is intended to increase the level of interoperability of German cancer data from syntactic to semantic one. The authors implement transformation of metadata and data conforming ADT/GEKID-Basis dataset (oBDS) into the OMOP Common Data Model (CDM).

The approach combines data preparation, automatic and manual mapping and ETL implementation. Large vocabularies and data models are carefully investigated and aligned. Comprehensive statistics of metadata and data mapping is provided.

The paper fits the topic of the special issue well. The work done is useful in the subject domain.

The main drawback of the paper is that it lacks methodological part and its style reminds a concise technical report. Automatic and manual mapping (2.3) as well as ETL (2.4) should be described in more structured and detailed way and accompanied by examples. Separate stages of mappings and their implementation should be distinguished as well as their input and output data.

It is very strange that dates “cannot be automatically or manually mapped”. It should be explained why with examples.

Discussion section should be structured in a better way. For instance, separate “lessons learned” should be distinguished, generalized, entitled and described to make them useful in a different subject domain with a similar complexity of vocabularies and data models.

Separate remarks:

·         [Introduction] Main results of the paper [lines 116-120] should be described in more details here. Structure of the paper should be described.

·         Conceptual issues of the approach should be separated from pure implementation issues (like PostgeSQL).

Minor remarks:

·         On https://www.gekid.de/ oBDS means “Basis dataset”, not “basic” as in the paper

·         [Figure 4] Font size is VERY small. Numbers are almost indistinguishable. Representation should be revised.

·         [Figure 3] “manuell” -> “manual”

·         [Supplementary Fig S2] dpi should be increased.

·         References 11, 15, 17, 20 are incomplete

·         For References 2, 3, 7 an English translation should be provided.

Author Response

(The authors gave the same response as above.)
